# A Miniature, Fiber-Optic Vibrometer for Measuring Unintended Acoustic Output of Active Hearing Implants during Magnetic Resonance Imaging

**DOI:** 10.3390/s21196589

**Published:** 2021-10-02

**Authors:** Guy Fierens, Joris Walraevens, Ronald Peeters, Nicolas Verhaert, Christ Glorieux

**Affiliations:** 1Laboratory of Soft Matter and Biophysics, Department of Physics and Astronomy, KU Leuven, Celestijnenlaan 200D, B-3001 Heverlee, Belgium; christ.glorieux@kuleuven.be; 2Cochlear Technology Centre, Schaliënhoevedreef 20I, B-2800 Mechelen, Belgium; jwalraevens@cochlear.com; 3Research Group Experimental Otorhinolaryngology, Department of Neurosciences, KU Leuven, Herestraat 49, B-3000 Leuven, Belgium; nicolas.verhaert@kuleuven.be; 4Department of Radiology, University Hospitals Leuven, Herestraat 49, B-3000 Leuven, Belgium; ronald.peeters@uzleuven.be; 5Department of Otorhinolaryngology, University Hospitals Leuven, Herestraat 49, B-3000 Leuven, Belgium

**Keywords:** vibrometer, acoustic implant, unintended stimulation, MRI safety

## Abstract

Making use of magnetic resonance imaging (MRI) for diagnostics on patients with implanted medical devices requires caution due to mutual interactions between the device and the electromagnetic fields used by the scanner that can cause a number of adverse events. The presented study offers a novel test method to quantify the risk of unintended output of acoustically stimulating hearing implants. The design and operating principle of an all-optical, MRI safe vibrometer is outlined, followed by an experimental verification of a prototype. Results obtained in an MRI environment indicate that the system can detect peak displacements down to 8 pm for audible frequencies. Feasibility testing was performed with an active middle ear implant that was exposed to several pulse sequences in a 1.5 Tesla MRI environment. Magnetic field induced actuator vibrations, measured during scanning, turned out to be equivalent to estimated sound pressure levels between 25 and 85 dB SPL, depending on the signal frequency. These sound pressure levels are situated well below ambient sound pressure levels generated by the MRI scanning process. The presented case study therefore indicates a limited risk of audible unintended output for the examined hearing implant during MRI.

## 1. Introduction

In medicine, magnetic resonance imaging (MRI) has become a commonly used technique for non-invasive diagnostics of many pathologies. The popularity of the technique is partly due to its lack of ionizing radiation and detailed soft tissue visualization, making it superior to computed tomography in many cases. Despite its popularity, MRI is not completely without risk and a number of adverse events have been reported over the years, including severe skin burns [1,2] and lethal accidents due to magnetically induced forces [3]. International industry organizations have therefore created a number of standards to regulate the use of MRI and limit patient risk as much as possible [4].

Additional risks arise when exposing a patient with an implanted medical device to an MRI environment as the, often conductive or metallic, implant and the scanner mutually interact, creating unwanted and even hazardous situations [4,5,6]. Over the past years, several severe adverse events have been reported with medical devices, including excessive local heating, unintended electrical stimulation, movement of implanted aneurysm clips or paralysis due to heating-induced brain lesions [3,7]. 

Both the number of implantations and the use of MRI are increasing rapidly, with in some cases a clear need for post-operative follow-up MRI [8,9] requiring that both technologies would be operated safely in parallel. Several innovations have been made by medical device manufacturers, trying to make their technologies safer for use in MRI, whilst standardization organizations have been working to regulate the combination of both technologies using a number of pre-clinical tests to limit the risks [10]. However, due to the large variety of devices, most standards are limited to covering generic risks and device-specific risks are not included in these standard works. 

One of these risks exists for active hearing implants that provide the patient with a mechanical form of stimulation, such as bone conduction or middle ear implants. These devices create an acoustic form of stimulation by either vibrating the skull or the middle ear of a patient [10]. Due to interactions with dynamic electromagnetic fields during MRI, these devices could unintentionally stimulate parts of the middle ear of a patient, leading to discomfort. This phenomenon has been reported with descriptions such as “loud bangs”, “continuous sounds”, or “flapping sounds when entering a scanner” [10,11,12,13]. As these interactions could potentially be harmful to the residual hearing of a patient, a method to quantify this form of unintended device output is required.

Unintended acoustic output of an implantable device would exist in addition to high ambient sound pressures that are always present when an MRI scanner is operating. These sound pressure levels are typically between 100–130 dBA for scanners with field strengths of 1.5 and 3 T [14,15,16] and contain frequency components up to 4 kHz [17,18] for most pulse sequences, with a peak around 1–2.3 kHz [15,19,20]. Due to the loud environmental sound pressure levels in combination with the expected duration of an examination, regulators require the use of hearing protection when MRI sound pressure levels exceed 99 dBA [21]. 

Measurement of unintended acoustic output requires a technique that is able to measure sub-nanometer movements [22] in the audible frequency range using a technique that does not suffer from electromagnetic interference. In otology, the output of acoustic implants or other acoustic energy sources is often quantified by measuring the vibration amplitude of certain anatomical structures using laser Doppler vibrometry (LDV) [23] or by measuring the complex pressure difference in the cochlear scalae using miniature fiber-optic pressure sensors [24,25,26]. Applying these techniques during MRI would, however, be challenging as the techniques would also be sensitive to the high ambient sound pressures present during scanning. Similar outcomes would be expected for optical microphones for active implantable hearing systems [22,27]. Commercial LDV systems have also been used to quantify vibrations of a patient bed during MRI scanning [28], where the laser beam was aimed at the measurement location from a large distance. The target site for measuring unintended output can be as small as 0.5 mm for middle ear implants [29] or 4 mm for bone conduction implants [30], making beam alignment from outside the 5 Gauss line practically unfeasible.

As no commercial sensors are available, a different approach to characterize the risk of unintended acoustic output of acoustic hearing implants is required. The current work first presents the design of a fiber-optic, MRI-safe vibrometer that is able to accurately measure any MRI-induced acoustic output. In the next section, the working principle of the vibrometer is demonstrated as well as its functionality during MRI. A scenario is investigated in which the vibrometer is used to quantify any unintentional output of the Cochlear™ Carina^®^ 2 (Cochlear Ltd., Sydney, Australia) fully implantable middle ear implant in a 1.5 Tesla (T) MRI scanner. This device features an implant body containing the electronics and a battery, a subcutaneous microphone and a miniature linear actuator that provides stimulation to the patient’s ossicles or inner ear. Prior research on this device has shown that no permanent displacement or actuator coupling are expected after 1.5 T MRI [31], yet to date there is no information for this device on any unintended acoustic output. The present paper aims to close the knowledge gap for this device. 

## 2. Materials and Methods

### 2.1. Principle of Operation

A schematic overview of the vibrometer is provided in Figure 1 below. The vibrometer uses a fiber-coupled light source that emits 660 nm light (IF-E97, Industrial Fiber Optics, Tempe, AZ, USA). The light is coupled into a 1-mm-diameter plastic optical fiber (980 µm simplex polyethylene optical fiber; Industrial Fiber Optics, Tempe, AZ, USA) that guides the light from the MRI control room into the scanner environment. The light is emitted from the polished end of the fiber into an air gap of 2 mm. At the opposite end of the air gap a second fiber is placed coaxially with the first fiber, capturing a part of the light transmitted through the air gap. A schematic overview is presented in Figure 2. Finally, the collected light is detected by a fiber-coupled photodarlington (IF-D93, Industrial Fiber Optics, Tempe, AZ, USA).

The light emitted from the fiber is assumed to follow a Gaussian distribution, where the irradiance I at the fiber exit can be described as:(1)$$I\left(x,y\right)=I_{0}e^{\frac{-2\left(x+y^{2}\right)}{w_{0}{}^{2}}}=\frac{2P_{0}}{\pi w_{0}^{2}}e^{\frac{-\left(x^{2}+y^{2}\right)}{w_{0}^{2}}},$$

With *I*_0_ the total output irradiance (W/m^2^), *P*_0_ the total output power (W), and *w*_0_ the 1/*e* beam width (mm). This assumption for multimode fibers is based on the work of Mawlud and Muhamad [32], an assumption which will be verified in a later stage. As the light is leaving the fiber via a small opening, it diverges when traveling through the air gap. A conical beam divergence can be assumed, creating the following linear relationship between the beam width and the distance with respect to the fiber exit:(2)$$w\left(z\right)=az+w_{0},$$

The presence of the air gap allows an object of interest to be placed between both fibers, shadowing part of the beam so that a vibration of said object in a direction perpendicular to the traveling direction of the light translates into a change in light power picked up by the receiving fiber. The relationship can be defined as:(3)$$P_{transmitted}=P_{tot}-P_{blocked},\;with$$
(4)$$P_{tot}=\int_{-\infty }^{+\infty }\int_{-\infty }^{+\infty }I\left(x,y\right)\;dxdy\;,$$
(5)$$P_{blocked}=\int_{-\infty }^{xT}\int_{-rT}^{rT}I\left(x,y\right)\;dxdy\;,$$
where *x_T_* and *y_T_* are the coordinate dimensions of the (here presumed rectangular) object of interest in the intersection plane perpendicular to the axes of both fibers (*x_T_* and *r_T_* are annotated in Figure 3b). Using Equations (3)–(5), the point of maximum sensitivity at a point z_1_ can be identified as the point where:(6)$$\frac{d}{dx_{T}}P_{transmitted}=\left|\frac{-dP_{blocked}}{dx_{T}}\right|,$$
(7)$$\frac{d}{dx_{T}}P_{transmitted}=\left|-\int_{-rT}^{rT}\frac{2P_{0}}{\pi w{\left(z_{1}\right)}^{2}}e^{\frac{-\left(x^{2}+y^{2}\right)}{w{\left(z_{1}\right)}^{2}}}\;dy\right|,$$

Is maximal or when *x_T_* equals 0 (Figure 3b). A simulation model of the described principle of operation was built in Matlab (MathWorks, Nattick, MA, USA).

### 2.2. Experimental Verification

The functionality of the vibrometer was verified in a three-step approach. In the first step, the assumptions made in the model were checked. The shape of the light beam exiting the polished fiber was captured using a single lens reflex camera (Canon, Tokyo, Japan) by placing the fiber in contact with paper to act as a transmissive medium (Figure 4). Attenuation losses were taken into account using the Beer–Lambert law. The assumption of conical beam divergence was then verified by capturing the beam shape at 0.5 mm increment positions in the *z*-axis. At each position, an image was acquired and saved in RAW format for further post-processing. The camera was set up with an ISO-value of 100 and a diaphragm size of 5.6 mm to capture as much of the light as possible. Based on the red values in the RGB matrix of each image, an intensity profile was determined that could be used to perform a Gaussian fit and determine the beam width.

In a second verification step, a custom 3D printed test setup was used, in which two pairs of airgap-separated fibers of 65 cm length each were mounted in parallel, identical air gap sizes (Figure 5). The first fiber pair was positioned so that the object of interest was partially obstructing the air gap, so that vibrations of the object resulted in variations of the transmitted light collected by the receiving fiber. The second fiber pair was positioned distally from the test object, with no obstruction of the respective optical path, and was used to characterize the noise level. The setup was used to characterize the electrical/optical system noise as well as the dynamic behavior of the vibrometer. The latter was achieved by positioning the (unloaded) tip of the Cochlear™ Carina^®^ 2 actuator (<0.5 mm diameter) in the middle between the two fibers and moving it through the light beam in steps of 20 µm using a linear translation stage (MTS50 with a KDC101 motor controller; Thorlabs GmbH, Newton, NJ, USA). A piece of retroreflective tape (A-RET-T010; Polytec GmbH, Irvine, CA, USA) with a 1-mm width was attached to the actuator to ensure that the light bundle could be completely blocked. Figure 3a shows a schematic overview of the test setup used to verify the vibrometer performance. At each step, the DC response of the photodarlington was recorded as a measure for P_tot_ and the dynamic sensitivity was measured by stimulating the actuator. Signals were acquired by an oscilloscope (TNS 1032B; Tektronix, Beaverton, OR, USA), while the actuator was stimulated using a UPV audio analyzer (Rohde and Schwarz, Munich, Germany) using a 1500 Hz sine wave of 0.5 V_RMS_. Noise signals were acquired for both channels afterwards by measuring 10 s sound fragments, using a soundcard operating at a 96 kHz sample rate (Fireface UC; RME, Haimhausen, Germany). Using these sound fragments, the correlation between both channels was calculated in order to investigate the reduction of the noise level in the channel of interest in case a common noise source is present in both signals.

Finally, the third step was used to verify the functionality of the vibrometer in the MRI environment. Experiments were performed in a 1.5 T Philips Achieva (Philips Healthcare, Eindhoven, The Netherlands) scanner in the University Hospitals Leuven. The vibrometer was placed in the scanner isocenter whilst the measurement equipment was located in the MRI control room using 9 m long fibers. Vibrations were created arbitrarily during scanning. All post-processing of data was done in Matlab.

### 2.3. Case Study

After verification of the vibrometer functionality, the custom 3D-printed test setup was used to expose the actuator and vibrometer to the MRI environment. As a preparation for the experiment, the actuator was mounted in the holder in the control room and the actuator functionality was verified by stimulating it with a stepped sine sweep for frequencies between 0.2 and 7.5 kHz. The noise level was also measured by recording a 5 s signal without stimulation. 

The actuator was then connected to the implant body of the Cochlear™ Carina^®^ 2 System before moving it into the MRI scanner. The implant was switched off before the experiment. Special care was given to avoid changing the relative positions of the fibers and the actuator. The parts were placed with their longitudinal axes aligned with the scanner bore axis (Figure 6, test position 1). The actuator was placed on a water-filled box sized 15 × 15 × 12 cm in order to provide the H-MRI scanner with a signal. In addition, the water container was placed on a 4-cm-thick polyurethane foam to dampen environmental vibrations. The actuator was put in the scanner isocenter, after which different pulse sequences were performed during which the vibrometer signals were recorded. Pulse sequences were selected to provide both high RF intensities as well as high gradient field intensities to consider both as potential contributors to unintended output. The key parameters for the selected pulse sequences are listed in Table 1. Full pulse sequence parameters are provided as Appendix A. The actuator was then moved to the edge of the bore to enlarge the contribution from the gradient field (Figure 6, test position 2). Finally, the actuator was moved to the side of the patient bed to further increase gradient field amplitude (Figure 6, test position 3).

The actuator was finally taken out of the scanner and placed back into the control room where the functionality of the actuator and setup were verified.

Next, the acquired data were post-processed to derive the equivalent sound level produced by the actuator during each scanning sequence. For each signal the frequency spectrum was calculated using a Welch estimate using a sliding Hamming window of 1000 samples, with 20% overlap between subsequent windows. Based on the actuator functionality measurement before the experiment a transfer function H was calculated to convert the measured voltage to a vibration velocity:(8)$$H\left(f\right)=\frac{V_{act}\left(f\right)}{A\left(f\right)},$$
where *A*(*f*) is the frequency spectrum in V/Hz of the actuator response and *V_act_*(*f*) is the known velocity profile of the actuator in mm/s for a specific voltage, as provided by the device manufacturer. The transfer function *H* was then used to convert the frequency spectra of the different acquired signals to velocity spectra. 

Assuming that any output of the actuator is transferred directly to the stapes footplate of a patient, an equivalent sound pressure can be calculated that would provide the same level of stapes vibration when acoustically stimulating the external ear:(9)$$p_{sig}\left(f\right)=\frac{V_{sig}\left(f\right)}{METF\left(f\right)},$$
where *p_sig_* is the pressure in the external ear canal in Pa that would lead to an identical vibration of the stapes footplate. The denominator in Equation (9) is the middle ear transfer function (*METF*) as defined in ASTM F2504 [33]. It represents the ratio of stapes movement in mm/s as a result of a sound pressure presented to the external ear canal in Pa. It needs to be noted however that Equation (9) makes two assumptions. The first one is that the detected vibrations are transferred directly to the stapes footplate. In reality, however, middle ear implants can be coupled to different parts of the middle ear structures, yet, mostly to the incus body [31]. The ossicular chain would still amplify these vibrations when conducting to the stapes. The latter would result in an underestimation of the resulting sound pressure if not for the second assumption, which assumes that the actuator transfers exactly the measured vibrations to the ossicular chain. However, in the used measurement setup the actuator was unloaded, leading to higher vibration amplitudes as opposed to when it would be coupled to the middle ear impedance. We assume that both effects partly cancel out and therefore provide a realistic estimate of the sound pressure and corresponding sound pressure level: (10)$$L_{E,sig}\left(f\right)=20*log_{10}\left(\frac{p_{sig}\left(f\right)}{2.10^{-5}\;Pa}\right),$$

## 3. Results

### 3.1. Simulation Model

Figure 7a shows the irradiance profile of the beam exiting the fiber as a function of the distance z to the fiber exit, assuming a conical beam divergence. This assumption is verified in the next paragraph. The 3D light profile is depicted by the plots in Figure 7b. 

Figure 8a shows the simulated amount of light transmitted through the airgap as a function of the x-coordinate (perpendicular to the beam axis) of the edge of the sheet that blocks part of the beam. The x-derivative of this curve is representative for the sensitivity of the transmitted light intensity to vibrations of the sheet (when connected to the actuator) and is visualized in Figure 8b. The sensitivity is clearly maximal when the sheet edge is halfway the beam, blocking half of its energy, as predicted by Equation (7). 

### 3.2. Experimental Verification

#### 3.2.1. Simulation Model Assumptions

The profile of the intensity across the centerline of the projection of the light spot on the camera-imaged paper, measured as described in Section 2.2, is shown in Figure 9 for several values of z. As expected, the irradiance profile follows a Gaussian distribution all along the beam. A Gaussian fit was performed on each dataset in Figure 9, thus yielding parameters a and w_0_ from Equation (2). A value of 0.38 ± 0.04 was found for a, whilst w_0_ was determined to be 0.24 ± 0.07 mm (Figure 10).

#### 3.2.2. Intensity Profile as a Function of Obstruction Distance

Gradually moving the retroreflective sheet through the air gap as a means of partially obstructing the light beam and thus converting changes tip position to changes of transmitted light power, while measuring the amount of light transmitted by a photodarlington transistor (Figure 11a), allowed us to measure the DC response of the system. The dynamic sensitivity of the vibrometer as a function of the obstruction distance was quantified by measuring the response of the photodarlington signal to 1500 Hz sinusoidal actuator oscillations (Figure 11b).

Both the dynamic and static response of the photodarlington change between approximately 0.6 and 1.6 mm, indicating that these are respectively the tip end locations where the light beam starts to be obstructed and is completely blocked. This implies that the experimentally observed most sensitive location is not in the theoretical center of the light beam: there is a difference of 100 µm between the expected center and the location of maximum sensitivity. This can be explained by a radial offset between both fibers, leading to a shift in the location of maximum sensitivity, as illustrated in Figure 12. In view of this, the simulation model was adapted to verify the effect of a radial offset and confirmed the hypothesis (Figure 13). 

Finally, the validity of Equation (6) was verified by numerically deriving the static response of the system and comparing that with the simulation prediction. The results of numerically deriving the three available datasets is shown in Figure 14 below. A discretization step of 0.02 was used in the derivation. Parameters of a Gaussian curve fitted to the data is provided in Table 2, together with the 95% confidence bounds on the calculated parameters.

As mentioned in Section 2.2, the setup consisted of two fiber pairs, both containing a gap between the input and output fiber, with one of the beams trespassing the gap being partially blocked by the actuator tip. In case a common noise source was present in the signals measured by both channels (e.g., due to ambient electromagnetic interference or optical noise), both signals would be highly correlated. This would allow noise reduction by subtraction. The correlation between both channels was determined by examining samples acquired by one channel in function of the samples acquired by the other channel (Figure 15). For illustrative purposes, a conversion was made between the acquired electrical response and the accompanying registered optical power by using the phototransistors sensitivity. A linear fit was performed on the data to investigate any correlation between both. This was carried out for four different datasets and results are summarized in Table 3 below. Datasets 1 and 2 were acquired before and after the MRI test on a first experimental campaign, while datasets 3 and 4 were acquired before and after MRI testing on a second campaign, which was organized to validate the first set of results. There is a negligible offset for all curves. However, for two datasets, the R^2^ value is close to one with a slope of 1, indicating a clear linear relationship and thus high correlation between both channels. The two other datasets show a less optimal fitting quality and slope diverging from 1, possibly due to a temporary source of noise in one of the channels.

#### 3.2.3. Functionality during MRI

Clear responses to dynamically changing complete blocking of one of the two beams could be observed when manually obstructing the light beam when the test setup was placed in an active MRI environment (Figure 16).

### 3.3. Optical Detection of MRI Induced Vibrations

Comparison between measurements of the actuator response before and after the experiment indicate that the device was still functional after MRI exposure. Further detailed device functionality testing was performed in the production environment of the manufacturer after the experiment. 

Spectra of optical transmission signals with the beam in one of both channels (the measurement channel) partially blocked by the actuator tip and the other one (the reference channel) uninterrupted were acquired during the different pulse sequences and with the setup positioned at different locations in and around the MRI scanner. Optically or electronically induced fluctuations of the measurement channel signal that were not related to the actuator movements of interest were suppressed by subtracting the signal from the reference channel. Figure 17 shows that the signal spectrum exhibits a peak in the 600–700 Hz range. For most locations, this peak exceeds with about 20 dB a background that slightly decreases with increasing frequency. In the isocenter of the MRI apparatus, the signal background is substantially higher compared to the other locations, especially towards lower frequencies.

In view of interpreting the sensitivity of the optical vibrometer, the measured spectra were converted into velocity values using the procedure outlined in Section 2.3. Tip velocities for all datasets are shown in Figure 18 below for both experimental occasions. By making use of Equation (9), the velocity spectrum was further converted into sound pressure level values. The shown 99 dBA level (black solid line) corresponds with the threshold used in industry standards for providing patients with hearing protection [21]. Results obtained 2 different measurement occasions are found to be consistent (Figure 19 and Figure 20).

## 4. Discussion and Conclusions

In the previous sections, an MRI-safe fiber-optic vibrometer was presented using 660 nm light in a robust and straightforward design. An approximate Gaussian light beam exited the end of a plastic optical fiber and diverged conically when traveling through an air gap before being partly captured by a second, coaxially located second plastic optical fiber. The initial assumption of a Gaussian light beam being emitted by the fiber was experimentally verified as illustrated in Figure 9 and Figure 10. From images acquired of the beam shape projected onto a piece of paper, the beam width could be derived. The uncertainty on the calculated beam width as a function of distance increase due to the spreading out of the energy over a larger surface, doing so increasing the uncertainty of the measurement and the accompanying fit. Bench testing confirmed the conical geometry of the divergence. Further testing verified the simulated dependence of the intensity of the collected light on the position of an actuator tip that was partially blocking the beam path. Also, the oscillation of the collected light power, which resulted from positional vibrations of the actuator tip, was examined. Comparison of the simulated static and dynamic behavior depicted in Figure 8 with the experimentally measured curves in Figure 11, shows that mainly the static behavior, or the DC response of the photodarlington has a steeper drop-off compared to the sigmoid-shaped curve predicted by the model. The dynamic behavior shows a narrower Gaussian shape compared to the simulated curve. Both deviations are likely a result of the actuator being positioned more closely to the fiber exit than assumed in the simulation, so that with changing position of the actuator tip, a larger fraction of the light beam was blocked in practice than assumed. Figure 11b indicates that the position with the highest sensitivity to positional changes of the actuator tip was not in the center of the light beam but had a 100 µm offset. This finding did not correspond with the expected result from Equation (7). This could be attributed to a radial offset between both optical fibers using the simulation model. 

A custom holder was fabricated that allowed mounting the sending and receiving fiber in a consistent way and accurately positioning the actuator in the light beam. A second pair of optical fibers was added to detect in parallel with the first pair possible signal contaminating noise, which could originate from environmental vibrations, electrical interference or any other factors. The noise level was characterized for both fiber pairs, and a correlation analysis was performed in order to examine the similarity between the noise induced fluctuations of the two signals, without any beam blocking. The measured collected intensity fluctuations show a good correlation for several acquired datasets, which confirms the feasibility of correcting the signal of interest by subtracting the signal acquired by the second, noise, channel.

Optical fibers of 9 m length were used to allow placing the holder in the MRI scanner whilst keeping all sensitive measurement equipment located in the EM shielded MRI control room. By manually obstructing the light beam during scanning, the functionality of the vibrometer was verified: there was no degradation in performance due to optical losses.

The system was exploited to verify the behavior of a Cochlear™ Carina^®^ 2 fully implantable middle ear system in an MRI environment. This apparatus has been labeled by the manufacturer as MRI unsafe, and to date it has been not clear whether there was a risk of unintended acoustic output of the device during MRI scanning. The Carina 2 system was taken off the market by the company in May 2020, yet there are a large number of patients implanted with this device that may have to undergo emergency MRI at some point in their lives. Unintended acoustic output of an active hearing implant has been reported in literature for similar devices [10,11,12,13], yet for this device no information was available. 

In order to investigate the risk of unintended output for the Carina 2 system, the middle ear actuator was used to partially block the light being transmitted through the air gap. The device was positioned at multiple locations in a 1.5 T cylindrical bore system to investigate the relative influence of the RF field and the gradient magnetic field during different pulse sequences at those respective locations. Pulse sequences were selected to provide high intensities of both fields and thus to create a worst-case environment. Spectra of signals acquired during scanning indicate very small signal amplitudes across the whole audible frequency range. 

Using operational data provided by the device manufacturer, combined with the measured actuator response when actively stimulating the device before the experiment, it was possible to convert the acquired signal spectra to tip velocity spectra. From the graphs in Figure 18 it is clear that vibrations can be detected down to 12 nm/s at 380 Hz, translating to peak displacements of minimally 8 pm at that signal frequency. The signal to noise of the system has feasible values up to about 10 kHz, limited by the conversion range provided in ASTM F2504 [33]. In addition to the velocity and displacement analysis, the mean middle ear transfer function, as defined in ASTM F2504 [33], was used to convert the tip velocity to an estimated sound pressure in the external ear canal that would lead to an identical vibration amplitude at the stapes footplate. Assumptions made during this conversion include a slight underestimation of the resulting vibration amplitude due to the additional amplification of the middle ear and a slight overestimation of the vibration amplitude due to the mechanical coupling of the device to the ossicles. We expect that both assumptions partly cancel out and that the results provided in Figure 19 and Figure 20 therefore provide a realistic estimate of the sound pressure level generated by the actuator during MRI. First and foremost, it needs to be noted that the curves acquired during MRI are located far below the reported system noise floor, similar to what was reported above. Secondly, all experimental curves are situated well below the desired floor level of 99 dBA as prescribed in IEC 60,601 Part 2-33 [21] for all frequencies. All spectra reveal a dominant frequency component present at approximately 672 Hz, including some harmonics at 1344 Hz and 2016 Hz. This could be a result of a mechanical resonance in the gradient coils, leading to high ambient vibrations or sound pressures [15].

Signals were acquired across two independent experimental campaigns, leading to a total of 26 datasets. These were acquired at different test locations, and for worst-case imaging sequences in terms of RF and gradient field amplitudes. For all datasets, equivalent sound pressure levels between 25 and 85 dB SPL, depending on the frequency, were deduced. All of these sound pressure levels are a lot smaller than the ambient sound pressure levels that were present during the scanning procedure itself. It is therefore highly unlikely that a patient implanted with a Cochlear Carina 2 system would experience discomforting acoustic stimuli from the implant during an MRI examination. Considering that hearing protection provided to normal hearing patients reduce the ambient level by roughly 20 dB SPL, it can be assumed that a normal hearing patient would be exposed to similar sound pressures as a patient with a Cochlear Carina 2 system.

The presented system offers an objective and accurate method to quantify the vibration of small components in an MRI environment. The MRI environment has been shown by many authors to be a hostile environment in terms of vibration [15,19] as well as in terms of sound level [15,17,18,19]. Quantifying miniature vibrations is therefore not straightforward. The designed system is able to cope with these factors by canceling out a large part of the signal noise by subtracting the signal from the second channel and by providing mechanical dampening of the test setup. It could be argued that an implantable optical interferometer as presented by Djinovic and colleagues [22,27] or fiber-optic pressure sensors implanted in the cochlea could allow more realistic in situ measurements. Despite the proven value of the latter techniques, they would be susceptible to the high ambient sound pressure levels, making it difficult to detect any unintended output. In the here-presented system, a number of assumptions have been made that were required to replace the in-situ measurement and avoid this issue.

In summary, this work presented the design of a novel, fiber-optic, and MRI-safe optical vibrometer that is able to quantify sub-micrometer movements of an object of interest, in this case: possible MRI-magnetic field-induced movements of the tip of an actuator. The functionality of this vibrometer concept was verified using a combination of simulation and bench testing. The sensor is therefore able to close the gap in commercial sensor technologies to quantify small amplitude vibrations in the MRI environment. A scenario was investigated involving a state-of-the-art fully implantable middle ear implant, showing that it is highly unlikely that a patient implanted with this device would experience audible stimulation during MRI. It is clear that the technique could be used for other medical devices as well. 

## Figures and Tables

**Figure 1 sensors-21-06589-f001:**
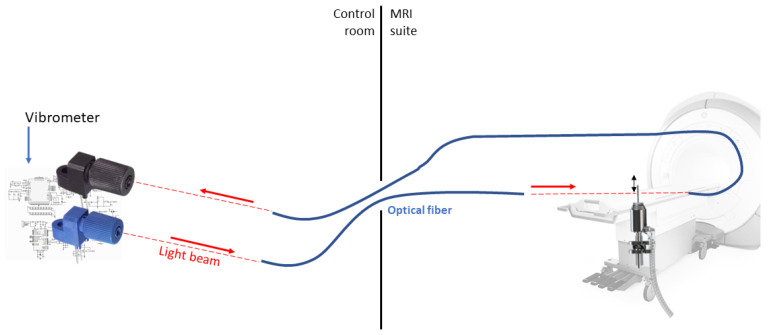
Schematic overview of the vibrometer (showing the different hardware components described above).

**Figure 2 sensors-21-06589-f002:**
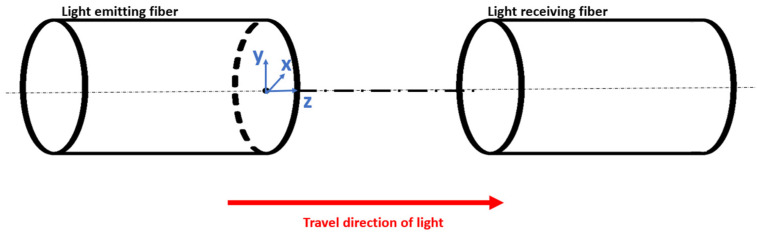
Schematic overview of the air gap (including coordinate system and direction of tip motion).

**Figure 3 sensors-21-06589-f003:**
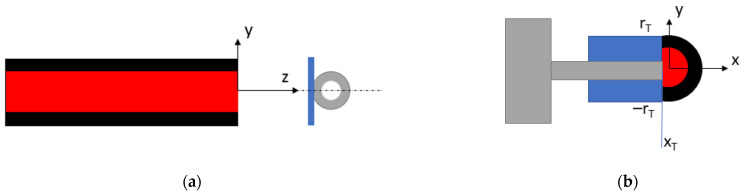
(**a**) Schematic overview of a longitudinal cross-section of verification test setup. The light-emitting fiber is visualized in red. The tip of the Carina middle ear actuator (gray) is placed perpendicular to the fiber axis to block the light traveling through the air gap. A piece of retroreflective tape (blue) is added to ensure that the light bundle could be blocked completely. (**b**) Schematic overview of a cross-section perpendicular to the fiber axis.

**Figure 4 sensors-21-06589-f004:**
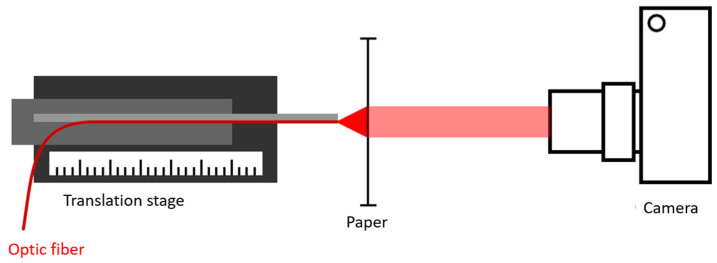
Schematic overview of the test setup used to quantify beam divergence.

**Figure 5 sensors-21-06589-f005:**
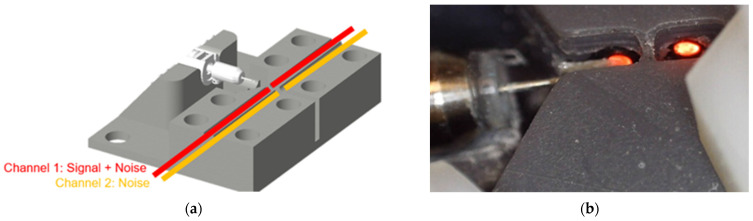
(**a**) Visualization of the 3D-printed test setup, showing the Carina middle ear actuator (white) and the two fiber pairs (red and orange) mounted in the setup (gray). (**b**) Close-up of the actuator positioned directly before the light beam (red).

**Figure 6 sensors-21-06589-f006:**
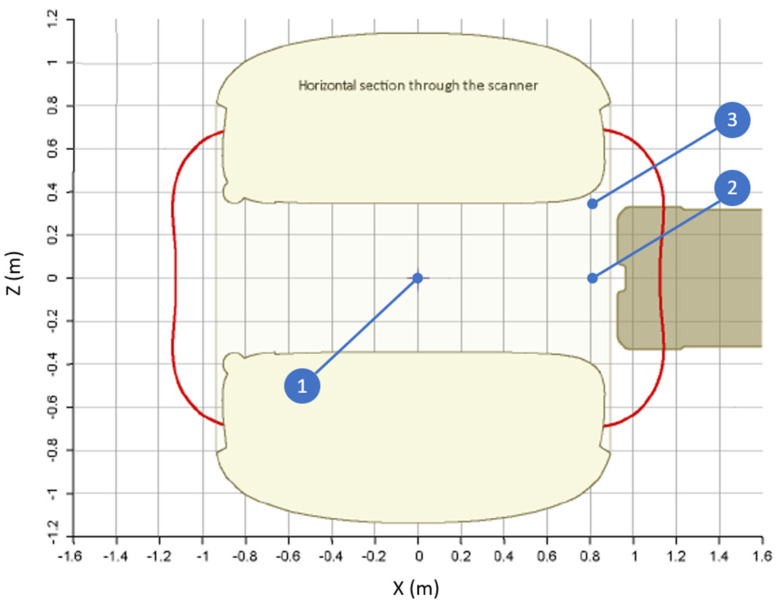
Schematic overview of the test positions in the scanner: (1) isocenter; (2) edge of the bore along the scanner axis; and (3) edge of the scanner bore and at the edge of the patient bed. Figure adapted from Philips Ingenia Technical Description (Philips Healthcare, Best, NL).

**Figure 7 sensors-21-06589-f007:**
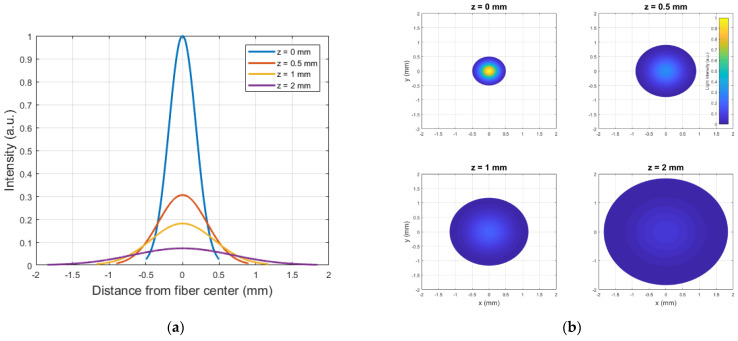
(**a**) Irradiance profile I (in a.u.) at a distance z from the fiber exit, considering conical beam divergence. (**b**) Irradiance profile (in a.u.) and beam with at a distance z from the fiber exit.

**Figure 8 sensors-21-06589-f008:**
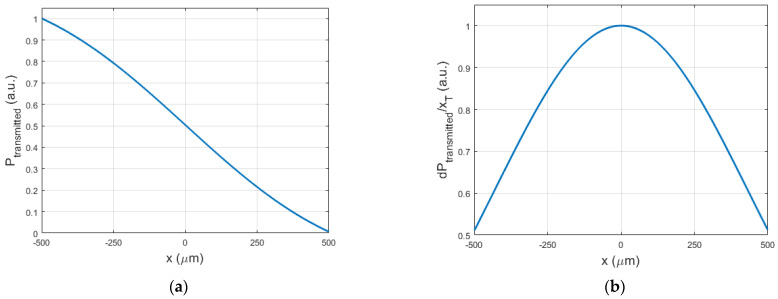
(**a**) Transmitted light power P_transmitted_ at z = 1 mm when the rectangular sheet blocks the airgap as a function of the radial beam coordinate x. (**b**) Spatial (x-) derivative of P_transmitted_ (x) 1 mm.

**Figure 9 sensors-21-06589-f009:**
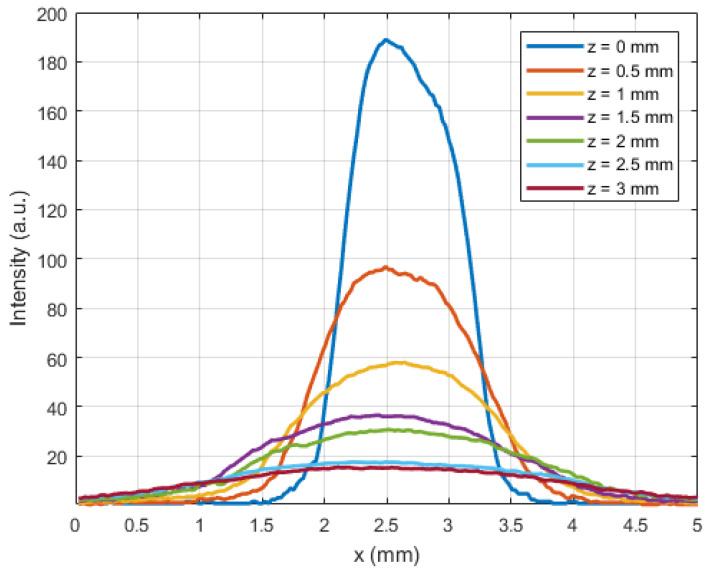
Light intensity profile measured along the x axis as a function of the axial beam coordinate z.

**Figure 10 sensors-21-06589-f010:**
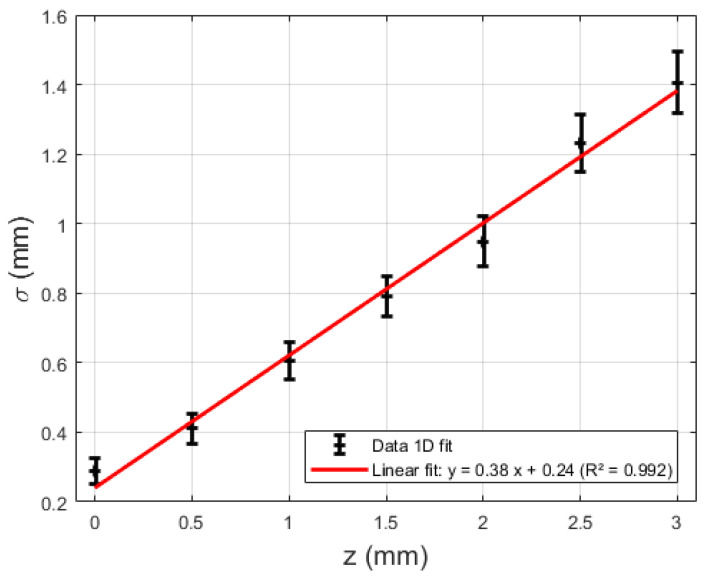
Beam with σ (mm) as a function of the axial beam coordinate z.

**Figure 11 sensors-21-06589-f011:**
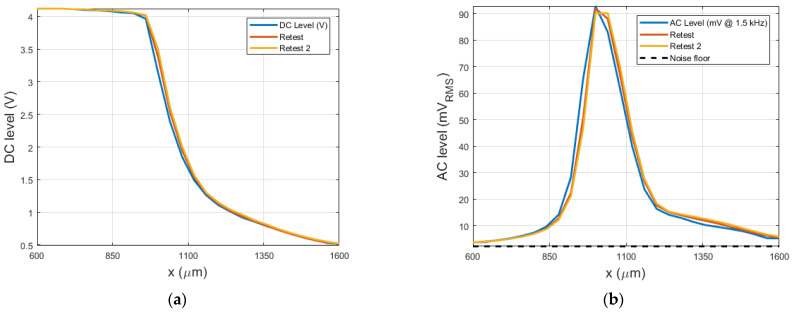
(**a**) Amount of light transmitted through the airgap as a function of the obstruction distance x. (**b**) Dynamic response when inducing a vibration using a Carina T2 middle ear actuator stimulated with a 1500 Hz 0.5 Vrms sine wave. Three datasets are shown, including the system noise floor (black, panel (**b**)) for the sensitivity curve.

**Figure 12 sensors-21-06589-f012:**
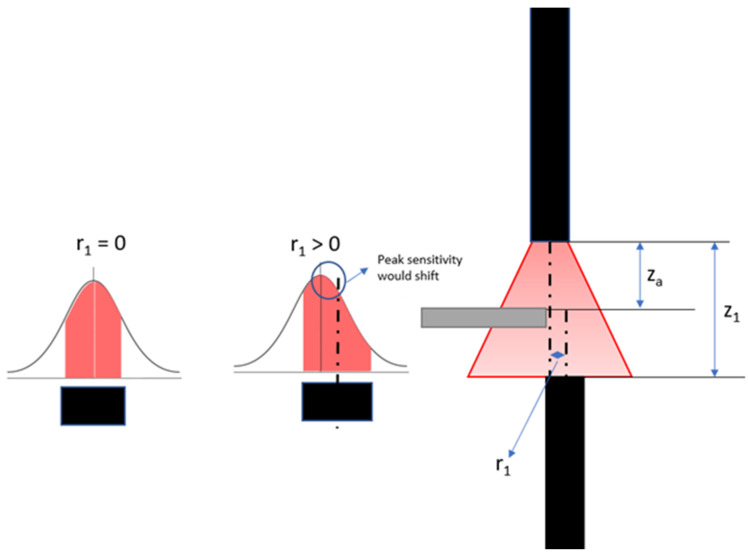
When a radial offset between both optic fibers is introduced, the location of maximum sensitivity shifts.

**Figure 13 sensors-21-06589-f013:**
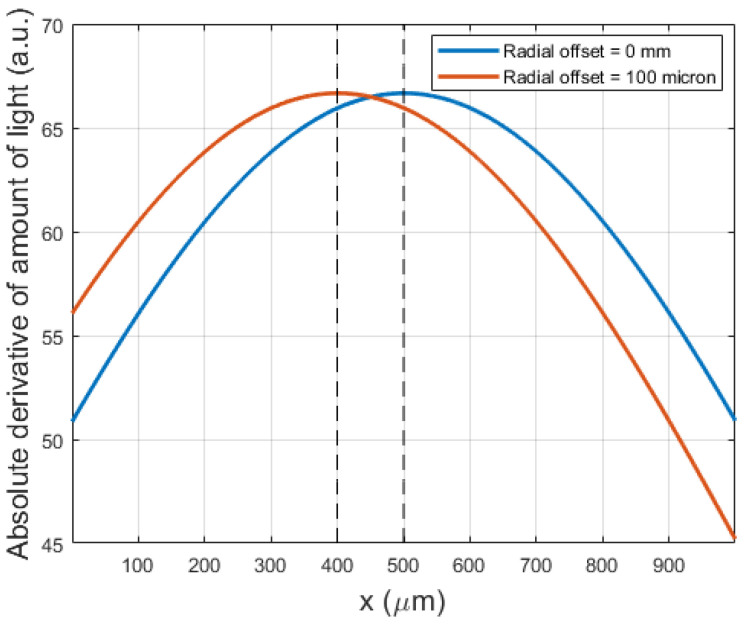
Simulation results of a radial offset, confirming the experimental findings.

**Figure 14 sensors-21-06589-f014:**
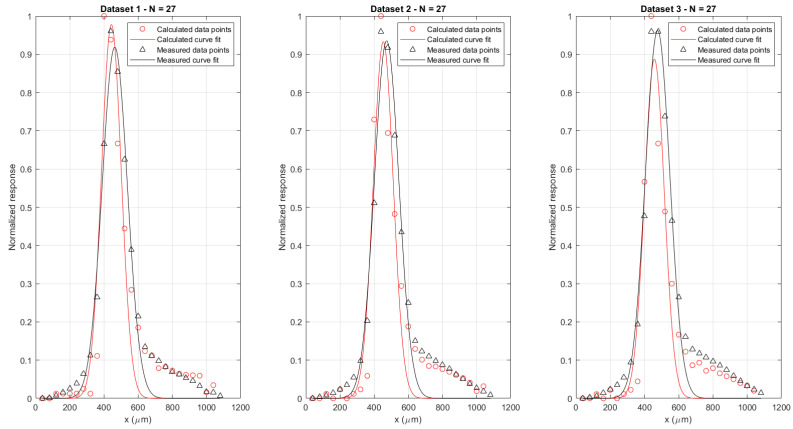
Comparison between the numerical spatial derivative of the static vibrometer response (red) and the measured dynamic response (black). Derivatives were calculated with a step size of 0.02.

**Figure 15 sensors-21-06589-f015:**
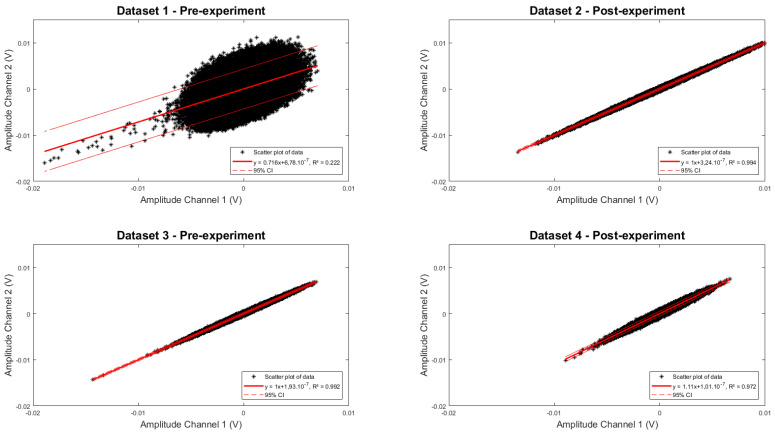
Correlation between signal amplitudes measured by channel 2 as a function of channel 1 (*n* = 48,000 samples). Channel identification as per Figure 4a. Four different panels are shown, illustrating the cross correlation for four different datasets.

**Figure 16 sensors-21-06589-f016:**
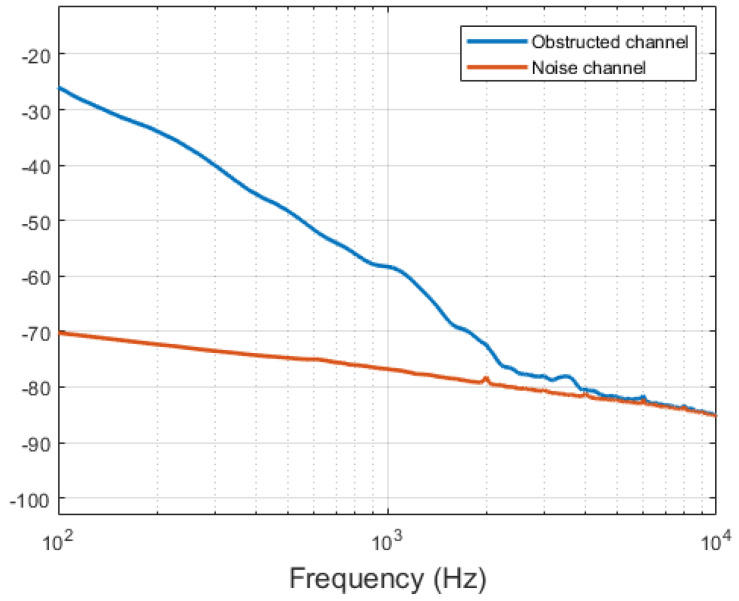
Amplitude spectra of signals acquired during MRI when one channel was manually partially obstructed (blue) versus the second, unaffected channel (red).

**Figure 17 sensors-21-06589-f017:**
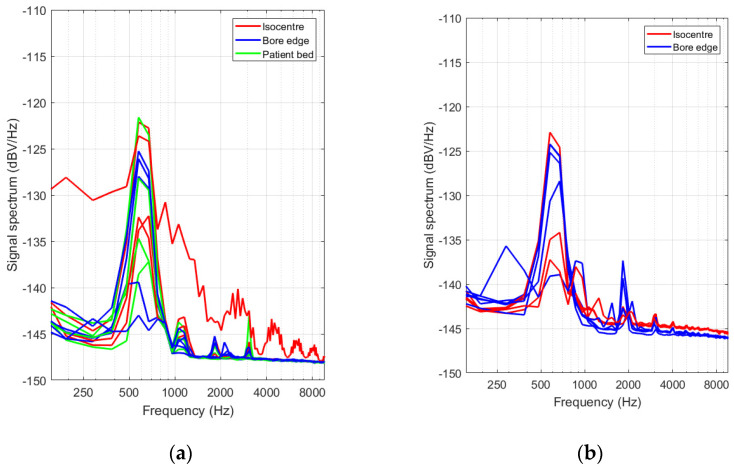
(**a**) Fiber transmission signal spectra acquired with the actuator tip partially interrupting the beam in the gap between the input and output fiber in one of the two channels. Different datasets are grouped according to their respective test location, indicated by different colors: scanner isocenter (red solid line), end of the bore on the patient bed (green) and right next to the scanner bore opening (blue). (**b**) Amplitude spectra acquired for signals acquired on the second experimental campaign. Color coding is identical as in panel (**a**).

**Figure 18 sensors-21-06589-f018:**
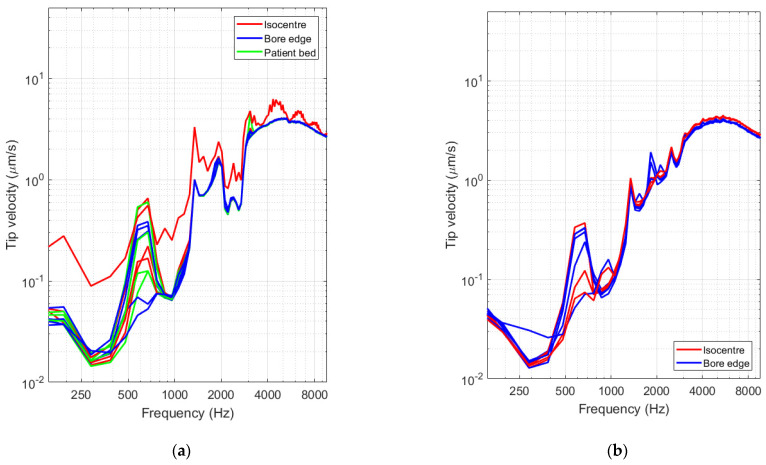
(**a**) Spectra of the signals in µm/s. Datasets are grouped per test location, showing signals when the actuator was located in the scanner isocenter (red solid line), at the end of the bore on the patient bed (green) and right next to the scanner bore opening (blue). (**b**) Spectra acquired for signals acquired on the second experimental campaign. Color coding is identical as in panel (**a**).

**Figure 19 sensors-21-06589-f019:**
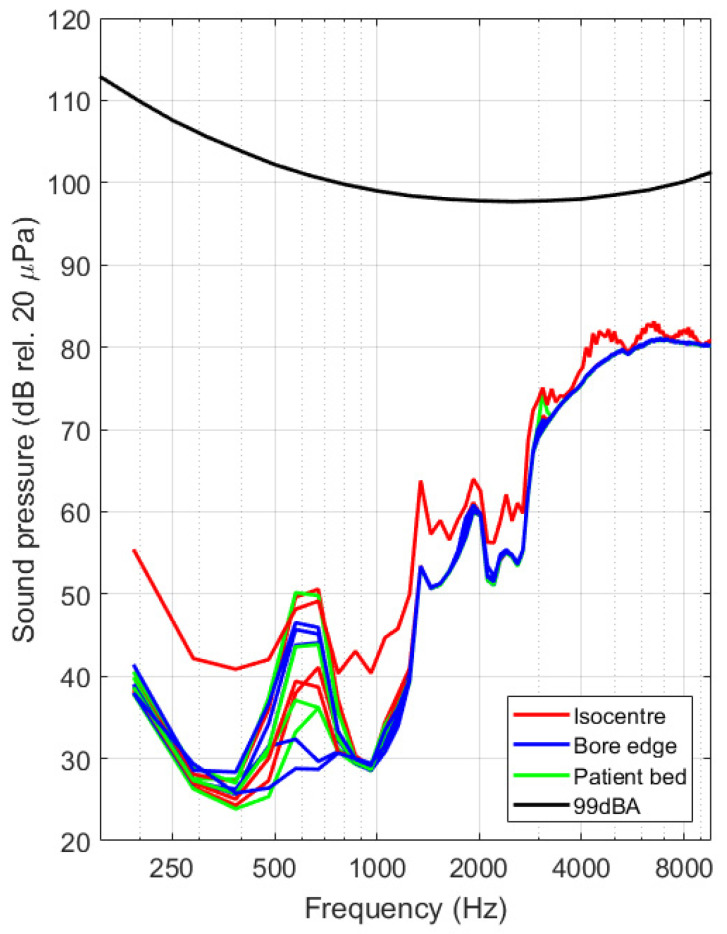
Spectra of estimated sound pressure levels for the acquired datasets, grouped per test location, showing signals when the actuator was located at the scanner isocenter (red solid line), at the end of the bore on the patient bed (green) and right next to the scanner bore opening (blue). The maximal acceptable noise floor of 99 dBA [21] is visualized using a black solid line.

**Figure 20 sensors-21-06589-f020:**
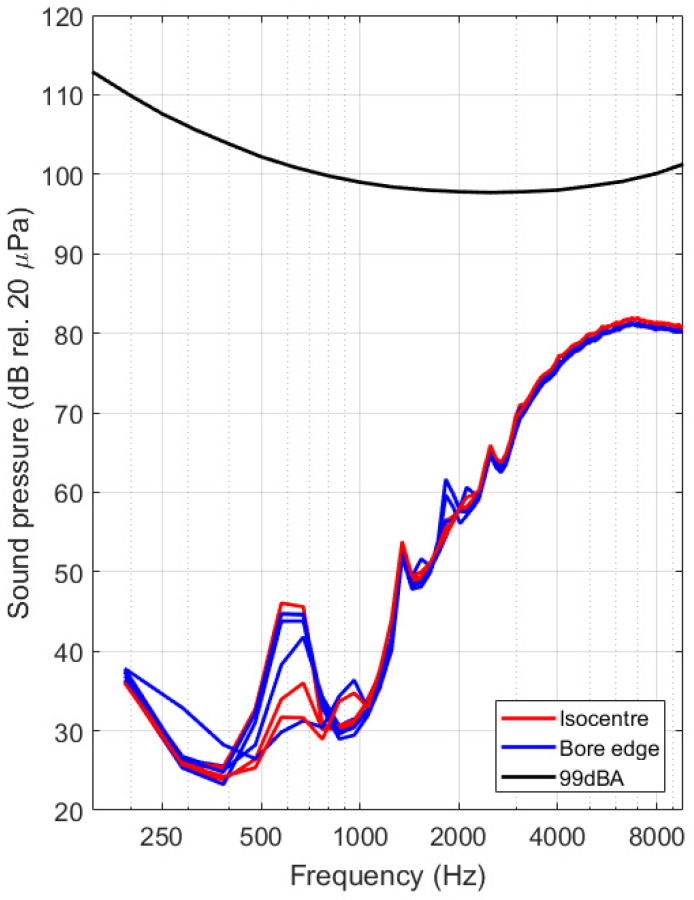
Spectra of estimated sound pressure levels for the datasets acquired during the second experimental campaign. Datasets are grouped per test location, showing signals when the actuator was located in the scanner isocenter (red solid line), at the end of the bore on the patient bed (green) and right next to the scanner bore opening (blue). The minimal acceptable noise floor of 99 dBA [21] is visualized using a black solid line.

**Table 1 sensors-21-06589-t001:** Key parameters for the pulse sequences that were used during the experiment.

Sequence Name	Gradient Field Strength (dB/dt)	RF Field Intensity (B1_+RMS_)
Diffusion weighted imaging (DWI)	62.4 T/s	1.47 µT
T2-weighted Turbo Spin Echo (TSE)	95.1 T/s	3.76 µT
Echo Planar Imaging (EPI)—Anteroposterior readout	112.6 T/s	0.92 µT
Echo Planar Imaging (EPI)—Inferosuperior readout	124.8 T/s	0.92 µT
Echo Planar Imaging (EPI)—Left–right readout	124.8 T/s	0.92 µT

**Table 2 sensors-21-06589-t002:** Parameters of Gaussian fits to the numerical spatial derivative of the static vibrometer response and the measured dynamic response. Coefficients are provided with 95% confidence bounds according to equation y = a x e^(((x − b)/c)^2)^.

Dataset	Parameter	Numerical Derivative	Dynamic Response
1	A	0.98 (0.81, 1.15)	0.92 (0.84, 0.99)
	B	443 (432, 455)	463 (456, 470)
	C	82 (65, 98)	104 (94, 115)
	R^2^	0.86	0.96
2	A	0.93 (0.79, 1.07)	0.94 (0.85, 1.02)
	B	454 (443, 464)	473 (465, 481)
	C	83 (68, 97)	102 (91, 113)
	R^2^	0.89	0.95
3	A	0.98 (0.85, 1.11)	0.96 (0.87, 1.06)
	B	478 (466, 490)	477 (469, 485)
	C	111 (94, 127)	102 (90, 113)
	R^2^	0.89	0.95

**Table 3 sensors-21-06589-t003:** Linear fit for the samples acquired by channel 1 in function of samples acquired by channel 2 in the form of y = ax + b. The coefficient of determination R^2^ is provided as a measure for the quality of the fit together with the 95% confidence intervals of the fitted coefficients.

Dataset	a (95% CI)	b (95% CI)	R^2^
1	0.9995 (0.9993–0.9998)	−2.10^−8^ (−6.10^−8^–2.10^−8^)	0.992
2	1.106 (1.105–1.106)	−1.10^−8^ (−8.10^−8^–6.10^−8^)	0.972
3	0.716 (0.712–0.719)	6.10^−8^ (−5.10^−7^–6.10^−7^)	0.222
4	1.001 (1.001–1.002)	−3.10^−8^ (−7.10^−8^–1.10^−8^)	0.994

## Data Availability

The data presented in this study are available on request from the corresponding author. The data are not publicly available due to the confidential nature of product manufacturing information.

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
