# Peer review of "A Miniature, Fiber-Optic Vibrometer for Measuring Unintended Acoustic Output of Active Hearing Implants during Magnetic Resonance Imaging"

_sensors, 2021, doi:10.3390/s21196589_

Round 1
Reviewer 1 Report
The manuscript entitled “A Miniature, Fiber-Optic Vibrometer for Measuring Unintended Acoustic Output of Active Hearing Implants During Magnetic Resonance Imaging” is interesting but some issues should be discussed.
- The Gaussian distribution of optical output is strictly true for single mode guidance. The multimode fibre output radiation profile (diameter 1mm) is strongly unstable and depended on the curvature of the fibre. Please comment it and add a reference to justify the Gaussian profile distribution in used fibre.
- Is necessary to add a noise reference fibre? Do the high frequency modulation of intensity can be used for noise cancelling?
- What is the relation of voltage presented in Fig. 15 vs the optical power?
- What is the maximum frequency of recording of electronic circuit (detector, electronics) ?
- What was used for input power of optical radiation stabilization?
Author Response
The manuscript entitled “A Miniature, Fiber-Optic Vibrometer for Measuring Unintended Acoustic Output of Active Hearing Implants During Magnetic Resonance Imaging” is interesting but some issues should be discussed.
Answer: We thank the reviewer for his/her time to review and provide feedback on our manuscript. The comments provided were addressed below as well as in the manuscript.
Point 1: The Gaussian distribution of optical output is strictly true for single mode guidance. The multimode fibre output radiation profile (diameter 1mm) is strongly unstable and depended on the curvature of the fibre. Please comment it and add a reference to justify the Gaussian profile distribution in used fibre.
Answer: The Gaussian profile distribution in the fiber was initially considered as an assumption when modelling the test setup. Later measurements did however confirm that this assumption was valid and that the emitted light beam was Gaussian. The results of these measurements are shown in paragraph 3.2.1 in the manuscript. Nonetheless, we have added a reference on the Gaussian profile distribution in lines 119 and 120 to increase the evidence level for this assumption.
Point 2: Is necessary to add a noise reference fibre? Do the high frequency modulation of intensity can be used for noise cancelling?
Answer: The purpose of the noise reference fiber is mainly to cope with vibrations of the entire vibrational setup. Initial, undocumented experiments indicated that if movement of the fiber holder shown in Figure 4a is created, the fibers could also move with respect to one another. The accompanying response can be considered as noise and needs to be removed. By adding the second fiber to the test setup, this noise source can be filtered out.
Point 3: What is the relation of voltage presented in Fig. 15 vs the optical power?
Answer: From an electric point-of-view, the circuits powering the LEDs feeding into both channels is identical. As the optical output power is directly related to the supply current of the LED, it is almost identical between both fibers. Secondly, the fibers for both channels follow the exact same physical path, so it can be assumed that the optical losses are similar. The measured electrical responses should therefore be representative for an almost identical optical power.
Point 4: What is the maximum frequency of recording of electronic circuit (detector, electronics)?
Answer: The maximum frequency is limited by the sampling frequency of the soundcard used to acquire the signals. The soundcard sampled at a frequency of 96 kHz, meaning that the maximum detectable frequency is 48 kHz.
Point 5: What was used for input power of optical radiation stabilization?
Answer: No power stabilization was performed for the documented test setup. None of the measurements that were performed during the entire development process indicated any signal drift, as the responses that were acquired were always reproducible and stable. We therefore expect that any kind of power fluctuations are negligible. In addition, vibrations on longer wavelengths that could cause fluctuations are expected to be identical for both measurement arms. By subtracting both channels, any influence would be cancelled out.
Reviewer 2 Report
Summary
The aim of this work was to design a fiber-optic vibrometer and experimental setup for measuring unintended acoustic output in a MRI environment. The setup was first validated with simulated data and test measured data, and then two repetitions of measurements before and after MRI scan were acquired and analyzed. The parallel two-channel design provided reference noise level to separate the signal of the unintended acoustic output from the environmental noise. A quantification analysis of the unintended acoustic output was conducted after converting to velocity and pressure. The manuscript was well structured and easy to follow the motivation and experimental setup. The illustrations made the manuscript easy to understand. There are still some minor issues need to be addressed.
Comments
Page 4, equation (5): Please define r and T in the manuscript.
Page 10, figure 10: Please discuss the standard deviation of sigma increasing as a function of z.
Page 13, figure 14: The highest points in the calculated data are all missed in the Gaussian fitting. Was there any outlier exclusion applied to the fitting? Could the authors explain why the measured data are wider than the calculated ones?
Page 14, figure 15: Please label these four subfigures for better understanding which dataset is which.
Although the authors explained the wide spread in the first subfigure was due to temporary source of noise, it is not clear why it happened and will it happen again and what negative impact it causes on the data analysis. Please discuss this in depth.
Page 16, figure 17: Why are the spectra at isocenter so different from that at bore edge? In addition, what made the results of the two repetitions so different ?
Page 19, line 443: Since the manufacturer labeled the device MRI unsafe, is this device MRI compatible after all? In other words, is the device magnetic and could eddy current be induced during MRI scans?
Author Response
Summary
The aim of this work was to design a fiber-optic vibrometer and experimental setup for measuring unintended acoustic output in an MRI environment. The setup was first validated with simulated data and test measured data, and then two repetitions of measurements before and after MRI scan were acquired and analyzed. The parallel two-channel design provided reference noise level to separate the signal of the unintended acoustic output from the environmental noise. A quantification analysis of the unintended acoustic output was conducted after converting to velocity and pressure. The manuscript was well structured and easy to follow the motivation and experimental setup. The illustrations made the manuscript easy to understand. There are still some minor issues need to be addressed.
Answer: We thank the reviewer for his/her time to review our work. The manuscript has been revised based on the provided comments. In addition, another check for the English language and style has been performed, in line with the provided feedback.
Comments
Point 1: Page 4, equation (5): Please define r and T in the manuscript.
Answer: Both xT and rT have been visually illustrated in Figure 5b. A reference to this figure has been added in the text below equation 5.
Point 2: Page 10, figure 10: Please discuss the standard deviation of sigma increasing as a function of z.
Answer: We have elaborated on the increasing standard deviation of sigma in the manuscript on lines 411-416. The increasing uncertainty of the fit can be attributed to the spreading out of the fiber, leading to lower intensity and less differing pixel values, thus impacting the certainty of the Gaussian fit.
Point 3: Page 13, figure 14: The highest points in the calculated data are all missed in the Gaussian fitting. Was there any outlier exclusion applied to the fitting? Could the authors explain why the measured data are wider than the calculated ones?
Answer: The Gaussian curves depicted in Figure 14 provide the best fit for the provided experimental/calculated data, minimizing the residual error. No exclusion criteria were used when either calculating the numerical derivative nor for the experimental data. The difference in width between the experimental/calculated datasets can be due to slight measurement errors in the experimental curves that were used. Voltages acquired during the experiment were sampled from a continuous time signal, which could be prone to small variations.
Point 4: Page 14, figure 15: Please label these four subfigures for better understanding which dataset is which.
Answer: Figure 15 has been adapted to contain labelling for the different subfigures.
Round 2
Reviewer 1 Report
The explanation for Point 3 is not valid in my opinion. The optical power vs. current or voltage response of the circuit depends on the detector characteristic and electrical circuit. If it is possible please verify the measurement using optical power meter.
Author Response
The explanation for Point 3 is not valid in my opinion. The optical power vs. current or voltage response of the circuit depends on the detector characteristic and electrical circuit. If it is possible please verify the measurement using optical power meter.
Answer: First of all, the authors like to thank the reviewer for taking the time to review our work for the second time. Secondly, we have resolved the unclarity with respect to Fig15’s relationship between the acquired voltage response and optical response by:
- Verifying the electrical characteristics of the vibrometer prototype. The response is measured as a voltage over a 1991 Ohm resistor that is in series with the phototransistor. Resistance values are acquired with 0.1 Ohm accuracy.
- Based on the datasheet of the IF-D93 phototransistor, the sensitivity at a wavelength of 660 nm is 5300 µA/µW. Using Ohm’s law in combination with the reported sensitivity, a voltage of 1V would correspond to an optical power of approximately 95 pW.
Using this information, Fig15 has been converted from an electrical response in Volt to an optical power response in micro-Watt. From the above, it is clear that the electrical characteristics are near to identical. A difference in resistance of the measurement accuracy would only result in a 0.005% error in the estimated optical power.
From an optical point-of-view, the fiber and its trajectory are also important with respect to the incoming optical power. As the fibers for both measurement channels are mechanically fixed to one another, it can be assumed that the trajectories for both fibers - and any influences due to the trajectory - are nearly identical.
Due to time constraints, we have not been able to do additional measurements using an optical power meter. However, the authors hope that the reviewer can agree with the presented evidence listed above regarding the optical power response for both channels.
